# *Wolbachia* Interactions with Diverse Insect Hosts: From Reproductive Modulations to Sustainable Pest Management Strategies

**DOI:** 10.3390/biology13030151

**Published:** 2024-02-27

**Authors:** Moazam Hyder, Abdul Mubeen Lodhi, Zhaohong Wang, Aslam Bukero, Jing Gao, Runqian Mao

**Affiliations:** 1Guangdong Key Laboratory of Animal Conservation and Resource Utilization, Guangdong Public Laboratory of Wild Animal Conservation and Utilization, Guangdong Engineering Research Center for Mineral Oil Pesticides, Institute of Zoology, Guangdong Academy of Sciences, Guangzhou 510260, China; sahito.2k10pt192@hotmail.com (M.H.); wangzh@giz.gd.cn (Z.W.); 2Department Plant Protection, Sindh Agriculture University, Tandojam 70080, Pakistan; amlodhi@sau.edu.pk; 3Department of Entomology, Sindh Agriculture University, Tandojam 70080, Pakistan; abukero@sau.edu.pk

**Keywords:** *Wolbachia*, insect hosts, reproductive modifications, disease vectors and pest control

## Abstract

**Simple Summary:**

The usefulness of *Wolbachia*-based control methods for several insect orders is investigated in this synthesis, with a particular emphasis on sterile insect technique (SIT) and incompatible insect technique (IIT). Strong control tactics and financing are required, as demonstrated by the integration of SIT into the management of *Drosophila suzukii* and the evaluation of a new SIT/IIT combination against *Aedes* mosquitoes. The potential of *Wolbachia* includes biological vector control in agriculture, particularly defence against pests that affect rice. Examining host-shifting dynamics and phenotypic impacts, especially in scale insects, highlights the ecological connectivity that is essential to comprehending the complexity of *Wolbachia*. The summary discusses the worldwide distribution of *Wolbachia*, focusing on genetics, medicinal uses, and mutualistic and parasitic adaptations. With developments in gene functional assays and multiomics, *Wolbachia* research is changing and becoming a paradigm for microbial symbiosis, with profound effects on education and translational research.

**Abstract:**

Effective in a variety of insect orders, including dipteran, lepidopteran, and hemipteran, *Wolbachia*-based control tactics are investigated, noting the importance of sterile and incompatible insect techniques. Encouraging approaches for controlling *Aedes* mosquitoes are necessary, as demonstrated by the evaluation of a new SIT/IIT combination and the incorporation of SIT into *Drosophila suzukii* management. For example, *Wolbachia* may protect plants from rice pests, demonstrating its potential for agricultural biological vector management. Maternal transmission and cytoplasmic incompatibility dynamics are explored, while *Wolbachia* phenotypic impacts on mosquito and rice pest management are examined. The importance of host evolutionary distance is emphasised in recent scale insect research that addresses host-shifting. Using greater information, a suggested method for comprehending *Wolbachia* host variations in various contexts emphasises ecological connectivity. Endosymbionts passed on maternally in nematodes and arthropods, *Wolbachia* are widely distributed around the world and have evolved both mutualistic and parasitic traits. *Wolbachia* is positioned as a paradigm for microbial symbiosis due to advancements in multiomics, gene functional assays, and its effect on human health. The challenges and opportunities facing *Wolbachia* research include scale issues, ecological implications, ethical conundrums, and the possibility of customising strains through genetic engineering. It is thought that cooperative efforts are required to include *Wolbachia*-based therapies into pest management techniques while ensuring responsible and sustainable ways.

## 1. Introduction

The vital symbiotic role of microbes in the eukaryote cell origin, their species conformation, ecological inter-relationship, and plant and animal activities around the biosphere are progressively evident [1,2]. The most widespread microbe in the animal kingdom, in which rules of engagement within species and obligations for research may shape, is the alpha-proteobacterium *Wolbachia pipientis,* which was recorded as a Rickettsia-like organism some hundred years ago in the gonads of numerous insects such as *Culex pipiens* [3]. The most prevalent intracellular bacterial species is *Wolbachia*, which can change the reproduction of host insects. Production of female infected progenies is enhanced through parthenogenesis, male killing, feminization, or cytoplasmic incongruity. These changes are considered adjusted for *Wolbachia* and promote their spread and multiplication [4,5]. Arboviruses such as Yellow fever virus, Zika and Chikungaunya are disease-causing agents that are disseminated through arthropods and are known to cause serious public health issues [6]. The worldwide occurrence of characterized arbovirus incidence has spiked in the recent few decades, emphasizing the demand for efficient control. Biological control is a promising option, wherein the application of *Wolbachia* surfaces is a potent option. Owing to its appreciable control potential against insects and viral pathogens, *Wolbachia* proved itself as a potential management strategy against insect vectors [7,8,9].

*Wolbachia*, one of the best heritable groups of endosymbionts, is extensively distributed among members of arthropods and a few nematodes [10,11,12]. They are the most diverse and abundantly found group of symbiont bacteria present on the globe: around 40% to 60% of the insect species are infected with different strains of *Wolbachia* [13,14]. During the last couple of decades, the most extensively studied aspects of *Wolbachia* involved their potential to trigger manipulations in reproduction [15] as well as their employment in the control of vector-borne diseases [16,17]. Identical to some other symbiotic organisms, the present phylogeny of *Wolbachia* is the result of three important processes: switching between the host species, diversification along with the host clade, and symbiont destruction [18,19]. *Wolbachia’s* dynamic localization and presence during *Drosophila oogenesis* emphasize complex interactions between the endosymbiont and host germline cells (Figure 1), offering important insights into the mechanisms controlling *Wolbachia* persistence and distribution throughout the egg’s developmental stages [4,20] as well as in a few strains invading bedbugs (super group F) [21]. Numerous studies could not provide evidence of mutual diversification between arthropods and strains of *Wolbachia* fig wasp [22], ants [23], butterflies [24] and bees [25].

Shifting of host is another hypothesis to justify the prevailing distribution of *Wolbachia* in dearth of co-diversification [26]. It shifts hosts, preferably through horizontal transfer when a strain of *Wolbachia* infects a new species of arthropod [27] and, at times, possibly through the process of hybridization [28,29]. It has been confirmed through various transmission studies that the event of host switching occurs by artificial inoculation of an uninfected species by a strain of *Wolbachia* [30], and the presence of “super predator strains” that have the potential to infect hosts distantly related in phylogeny (e.g., ST41 strain type in Lepidoptera) [31]. The transmission of *Wolbachia* physically to an uninfected host from the infected is the initial step during host switching, accomplished through different means of transmission, and is mostly assisted through biological vectors or an appropriate environmental factor [32,33]. The various means of transmission reported until date include the interaction of host parasites [33,34,35], the interaction of predator prey [36], and usual food sharing [37].

The interaction of a number of factors governs the population density of *Wolbachia* within a host population. These factors involve the sex, development stages, species involved, host genotypes, environmental factors, and strains of *Wolbachia* [38,39,40]. It is therefore very important to characterize the population density of *Wolbachia* and related factors to have an insight into the interaction between the symbiont bacteria and their hosts; the data so collected might also facilitate the standardization of management. The species of leaf hopper *Yamatotettix flavovittatus* is known as the vector of a phytoplasma disease of sugarcane called white leaf disease [41]. It is one of the most serious concerns about the sugarcane crop, causing appreciable yield destruction in Southeast Asian sugarcane producing countries [42,43]. Infections of *Wolbachia* with a very high frequency (>80%) have been recorded in the leafhopper natural population [44]. Furthermore, perfect vertical transfer and incompatibility in reproduction resulted from *Wolbachia* infections in leaf hoppers [45].

Parasitism of the reproductive system is the most widely reported impact of *Wolbachia* in arthropods, involving the alteration of reproduction in hosts that favour the persistence and transmission of symbionts generally through the increase in the relative population of bacterium in females of infected and uninfected hosts. Among the sexes, females typically transfer *Wolbachia* and the related heritable bacteria, though exceptions may exist but are very rare [46,47,48]. All four types of alteration of reproductive systems are involved in the case of *Wolbachia* [15,49,50]. In orders Lepidoptera, Isopoda and Hemipteran, genetic males, through the process of Feminization, convert into functionally active females. Parthenogenesis is triggered by *Wolbachia* in haplodiploid arthropod hosts such as Hymenoptera, Thysanoptera and Acari, in which females are produced from unfertilized eggs that were supposed to develop into males. To favour surviving female siblings infected with bacteria, the infected males occur during Male killing and are reported in Diptera, Pseudoscorpionida, Lepidoptera and Coleoptera. In cytoplasmic incompatibility (CI) [51], infected males are prevented from producing viable progenies on mating with uninfected females (missing *Wolbachia* or its compatible strains), which is believed to be the most widely occurring reproductive manipulation triggered by *Wolbachia* and is reported in Coleoptera, Isopoda, Acari, Hemiptera, Hymenoptera, Diptera, Orthoptera and Lepidoptera. The purpose of this review is to improve our understanding of *Wolbachia* ecological dynamics and its use in insect population control by highlighting important research gaps and outlining future initiatives. In order to ensure a more thorough knowledge of *Wolbachia* involvement across a variety of insect hosts, we suggest possible topics for inquiry and emphasise the necessity for joint efforts to overcome problems and develop techniques.

## 2. *Wolbachia* Biology: Host Ecology, Diversity, and Genomics

Alphaproteobacterium *Wolbachia* is a vital member of symbiotic microbes throughout the globe [1,2]. *Wolbachia*, a class of symbiont bacteria, is subdivided into various super groups A, B, E, H, and F with a variety of associations with host vectors, like *WPip* in *Culex pipiens*, *WalbA*/*WalbB* in Aedes albopictus, and Wmel in *Drosophila melanogaster* [52,53,54]. Various species of *Wolbachia* such as *WRi*, *WPip*, *WBm*, and *wMel*, have had their morphologies clarified during the genetic investigations [55]. Appreciable control has been revealed against RNA viruses by some of these strains, whereas few others had impacted the mosquito’s lifespan. These distinctions are related to environmental features and adaptability to hosts [56,57]. The genome of *WMelPop* is worth mentioning on account of its CG-rich sequence bias, rendering it highly pathogenic. Reproductive parasitism of *Wolbachia* relies on its intimacy in cytoplasmic incompatibility, which involves unidirectional as well as bidirectional mechanisms [58]. Intrinsic genes of immunity and those related to structural, metabolic and stress-linked mechanisms are also regulated by the bacteria [59]. Cases have been observed of both horizontal and vertical transmission of host reproductive alterations that involve feminization, male death, parthenogenesis, and cytoplasmic incompatibility [60,61,62,63,64]. Convincing results were revealed by *Wolbachia* in managing arthropod vectors. However, its impact on the native confrontation among males should also be considered [65]. *Wolbachia* has a vital ecological relationship with the host due to its eclectic effects on complex genetic features, reproductive alterations, and hosts. Its effective application in the management of disease vectors and other areas is emphasized due to all these important features.

*Wolbachia* is a universal facultative symbiotic bacterium that plays a vital role in arthropods and insects within the intricate domain of bacteria transmitted maternally. These inherent symbionts, commonly invading arthropods, and insects, are vital in tailoring the ecology of the host. They are highly dependent for their transmission and survival on hosts when transmitted vertically (Table 1), with affiliations ranging from obligate (concerning nutrition) to facultative (involving mutualism in parasitic relationships) [14,66,67]. In this domain, *Wolbachia* enjoys the key position as the most widely occurring facultative symbiont for a period as long as 200 million years ago [68]. Genus Alphaproteobacteria, a diverse group of intracellular G-ve bacteria, belongs to the order Rickettsiales and has affiliations with filarial nematodes and arthropods. *Wolbachia* has small, unique spherical cells, occupies testes and ovaries predominantly, and is associated with the female germline [69,70]. It has a ubiquitous presence in the insect species infecting 40 to 66% of the population with a range of prevalence and infection over time and space within insect populations [13,71,72]. The multiplex interrelationship that is present between *Wolbachia* and insect hosts spotlights the intrinsic symbiotic relationships that have repercussions both for the symbionts as well as for arthropod hosts under various ecological environments.

Genus *Wolbachia,* belonging to the family Rickettsaceae of the phylum Alphaproteobacteria, was identified for the first time in a mosquito (*Culex pipiens*) by Hertig and *Wolbachia* in 1924. Although initially there were many named taxa, *Wolbachia* primarily attributes to *Wolbachia* pipientis. Specific genome (DNA) sequences are targeted for recognition of the different isolates of *Wolbachia,* such as 16S rDNA, ftsZ, wsp and groEl [85]. Employing these genes, isolates were categorized into eight super groups (A, B, C, D, E, F, G, H), with each group having a unique host proclivity [86]. One must be very cautious while interpreting the phylogeny between *Wolbachia* isolates, taking into account the potential recombination and gene selection [87,88,89].

Initially, the *Wolbachia*’s genetic diversity was conducted based on the 16S rRNA gene [90] as well as wsp (the variable surface protein gene) [86]. As a result of extensive recombination in the wsp gene and the challenges faced in strains resolution and phylogenies due to reduced pace of 16S rRNA evolutionary rate, the MLST (multilocus sequence typing) system was introduced and established, involving housekeeping conserved genes [91]. All reported strains of *Wolbachia* were classified into at least 17 different (A–F, H–Q, and S) phylogenetic super groups, with most of the sequences from A and B super group strains [92]. With latest developments in the target enrichment protocols we are still in the phase of having a good insight on the genetic diversity [93,94]. The genetic landscape of *Wolbachia* is very complicated, as it is divided into various super groups based on the DNA sequences of few genes. To this group, the primary representative is *Wolbachia pipientis*. A two-tier method has been suggested to improve the clarity of phylogeny, exploring the evolution of *Wolbachia,* and to figure out the complicated relationships with reproduction in arthropod host. The present genetic research work being conducted on *Wolbachia* is helping to gain a clear understanding of the diversity of symbionts among the host arthropod species.

The intrinsic categorization of *Wolbachia* into fourteen super groups (A to O) is based on molecular characterization of important genes, such as multilocus sequence typing (MLST) loci, 16S rRNA gene, and *wsp* (surface protein gene) of *Wolbachia* [91]. Among the fourteen classified super groups (A to O) across different host taxa, most of the arthropod-related strains are in super groups A and B, nematode-associated strains are grouped in super groups C and D, whereas the rest of the super groups were observed infecting diverse hosts [95]. The latest developments in technologies employed for sequencing the DNA have facilitated the complete genome sequencing of *Wolbachia* strains with the complete sequence of *wMel* strain infecting *Drosophila melanogaster* to start with [96]. Strain *wMel* belongs to the super group A and has disclosed as little as 1.3 Mb (mega base) genome littered with free-moving genetic components.

Further insight on the host biology influencing genes and differences specific to strains was provided on subsequent sequencing of strains *wAlbB* of *Aedes albopictus* [97], *wRi* of *Drosophila simulans* [98], wPip of *Culex pipiens* [60] and wBm of *Brugia malayi* [99,100,101,102]. The data bank now has the genome sequence of more than one hundred genomes of *Wolbachia* infecting a variety of nematode and arthropod species [103]. Importantly, the developments in the field of DNA sequencing resulted in a change in the research field of *Wolbachia* that has unearthed complete genomes of many strains, including the *wMel* groundbreaking strain infecting *D. Melanogaster*. The research findings have laid the ground for comprehensive analysis that provides detailed insight into each specific strain, key genes that affect the symbiont–host interactions, and the complicated genome picture that is the basis of coevolution of *Wolbachia,* along with different nematode and arthropod hosts.

## 3. *Wolbachia* for Nature’s Collaborations to Control Insect Populations

In quest for innovative control strategies, *Wolbachia* proved to be an efficient as a biocontrol of mosquitoes, as it not only degrades the competency of vector but also exhibits appreciable amount of maternal transmission in various species like *St. aegypti*. The promising occurrence of the *wAlbB* strain in *St. aegypti* was observed in the case of embryo cytoplasm transfer from related mosquito specie (*St. albopicta*) [104]. Later, trans-infection involving *wMel* and *wMelPop* strains infecting *D. melanogaster* drastically reduced the *St. aegypti*’s competency for DENV during a laboratory assay [105,106,107]. The appreciable impact of *Wolbachia* stretches further to inhibit DENV in saliva of mosquitos, hence revealing its potential as a biocontrol agent. It is obligatory to invade the population of wild mosquitoes for efficient biocontrol. The infected females must transmit *Wolbachia* vertically (Table 1) at a very high rate to the offspring. Incredibly, all three strains of *Wolbachia* viz *wMel*, *wAlbB*, and *wMelPop* display a very high frequency of female transmission (close to 100%), and trigger increased CI (cytoplasmic incompatibility) (Table 2) in *St. aegypti* [104,105].

The increasing risk of vector-transmitted diseases demands significant investments in devising the alternative means of control. Biological control is among the oldest approaches recorded, ever since the ancient China, having the potential to address concurrent challenges. Deciphering the coordination between biocontrol and population cognizance, collective approaches appears more efficient in managing vector-transmitted diseases [122,123]. *Wolbachia*, particularly, protrudes in the management of *Aedes* mosquito population, extending a viable solution to the proliferating threat of arboviruses. In this context, the use of a chemical, such as larvicide, provides a meaningful solution to control mosquitos during the initial phase when young, but has limitations while combating the adults at the stage during which the vectors actually transmit virus. *Wolbachia* surfaces as an environmentally friendly and efficient substitute to control mosquitoes. The invaded mosquitoes not only furnish the management of arbovirus but also play a key role in controlling other similarly transmitted pathologies, and hence are instrumental for public heath safety [111,124].

The emerging realm of *Wolbachia* research, specifically against dengue and other related arboviruses, has potential for extended applications to manage different types of pathologies. Microinjection and transinfection are the two primary techniques used that operate as the integral tools. Trans-infection is the simple mechanical transfer of the symbiont however; micro-injection needs a qualified skilled person and also is costly. It is worth mentioning that the strains employed during this process are either extracted from eggs or procured from cell culture [30]. The most widely employed strain is *wMel* that presents appreciable amount of viability for the field release. *wMelPop* a transient strain of *Wolbachia*, has the potential to lowers the fecundity and viability of females of *Aedes* and hence reduce the population density in a locality. In addition, it has been revealed that strains of symbionts also restrict the multiplication of virus within the host, thereby further enhancing their efficacy as efficient biocontrol [109,125]. The vector competence of DENV is reduced when *Wolbachia*, applied as biological control agent, efficiently attacks St. aegypti. The strains of *Wolbachia* (*wMel*, *wAlbB* and *wMelPop*) transinfected result in cytoplasmic incompatibility and exhibit appreciable amount of maternal transmission. No matter when the vector-transmitted diseases are getting a big menace, *Wolabachia* offers an efficient and environmentally friendly tool of vector control. This, along with conventional biocontrol strategies, enhances the efficiency of disease management. Their methods of application such as transinfection have exhibited inspiring potentials for extensive pathogen management.

## 4. *Wolbachia* Dynamics in Lepidopteran and Dipteran Insects: Geographic Structure and Male Killing Aspects

The order of Lepidoptera contains many pests of agricultural significance besides containing modal species across the disciplines of biology [126]. Moreover, larvae of lepidopteran also act as hosts for parasitic wasps and flies [127,128]. It is a diverse group of insects and many of these have affiliations with other organisms however, there is very little know how about the bacterial populations affiliated with the members of order Lepidoptera. There is evidence of having mutualistic affiliations yet it is established that the *Wolbachia* has parasitic association with its arthropod host’s reproductive system [15]. It has both mutualistic as well as parasitic relationships with Lepidoptera hosts. Cytoplasmic incompatibility, androcide and feminization are the well established impacts of *Wolbachia* on Lepidopteran’s hosts reproductive system [129,130,131,132,133,134]. It has been reported that a species of *Wolbachia* is known to increase susceptibility of a lepidopteran host against baculoviruses, depicting it as an efficient biocontrol agent against *Spodoptera exempta*. The present study revealed an infection of a high proportion of Lepidopteran pests with *Wolbachia*, exhibiting the potential of *Wolbachia* in butterfly and moth biology [15].

It has been reported in an extensive survey that among 300 reported species of Lepidopteran, 43% were found invaded with *Wolbachia*. The reported figures noticeably exceed the previous regional reports of 16.2% infection frequency reported from Panama (*n* = 43 species [135]), 14.3% in the US (*n* = 21 [136]), 35.2% in the UK (*n* = 34 [137]), 45% in Japan (*n* = 49 [138]), 17% in Uganda (*n* = 24 [139]), 52% in India (*n* = 56 [140], and 43% in Western Europe (*n* = 7 [141]). It has been reported during a study of prevalence and distribution across the species that huge population of Lepidoptera are invaded with *Wolbachia* (approx. 80%), which is much higher than the estimated frequency across the insects when taken together [14,67]. Globally, there is an appreciable correlation between the host geography and the frequency of *Wolbachia’s* infection. The infection rate is usually high at lesser latitudes, depicting a tendency for warm temperatures. These findings were in contrast to some previous studies, where varied infection rates were recorded with no clear-cut correlation to the climatic conditions [142,143,144,145]. There was evidence that with an increased temperature, there was a reduction in infections caused by *Wolbachia* [146,147,148,149]. A significant prevalence of *Wolbachia* has been revealed in Lepidoptera during global investigations, defying preliminary findings. Additionally, besides many impacts of *Wolbachia* on Lepidoptera, the prevalent frequency of symbiont and geography of the arthropod hosts emphasises the complexity of the correlation and demands a dire need for further research to have a comprehensive insight of its subtleness.

Higher number of female infected offspring is produced as a result of infection due to *Wolbachia* through the process of feminization, parthenogenesis, male killing, or cytoplasmic incompatibility. Theses reported alterations are believed to be adaptive that promote the population of *Wolbachia* [4,5]. The elements of *Wolbachia* that were involved in cytoplasmic incompatibility of *Drosophila melanogaster* have been characterized recently However, the part of the genome responsible for governing the other three alterations is still unknown. The latest comparative genomics strategy recognized a candidate gene involved in male killing, which was found conserved in various male-killing strains of *Wolbachia*. The putative gene wmk (WO-mediated killing) encodes a presumed transcription factor having two DNA helix-turn-helix binding domains and is present within the eukaryotic affiliation module of *Wolbachia* prophage WO in numerous strains of *Wolbachia* so are the CI genes cifA as well as cifB [150,151]. The expression of *wmk* (codon optimized) trans-genetically in *D. melanogaster* triggers a number of cytological changes that ultimately results in the death of an embryo and also induces an appreciable female bias (average sex ratio of male: female is 0.65:1).

A species-specific and ecologically friendly method for controlling or eliminating pest populations is the Sterile Insect Technique (SIT), as seen in (Figure 2). Using this technique, sterile insects that are mass-produced are repeatedly released. Although successful deployments of both sexes have been reported, the majority of the insects released are males. Ionising radiation is the main method used to sterilise sperm, causing dominantly fatal mutations. To put it briefly, the SIT entails mass raising of the target species, releasing males (and maybe females) into the target region after separating and sterilising them. The treatment of *D. suzukii* is based on the SIT principles, which are illustrated in (Figure 2).

We thoroughly examined the tenets of the Incompatible IIT and SIT, investigating their possible application in the control of *Drosophila suzukii*. To successfully decrease the *D. suzukii* populations, we suggest including SIT as a crucial component of an area-wide integrated pest-control plan, taking into account the limitations posed by SIT and IIT [152]. Male killing has also been observed quite often in Lepidopterans (moths, butterflies, fruit flies). A very novel type of *Wolbachia*-triggered male killing has been reported in *Ostrinia scapulalis* and its related specie *O. furnacalis* [153,154], which is quite different from the male killings observed in other types of moths and butterflies [155,156]. The killing of *Wolbachia* by application of antibiotics resulted in all male progenies [153,154], revealing that the *Wolbachia* possessed the factor responsible for inhibiting masculinization or feminization in Ostrinia. Furthermore, the feminization factor in Ostrinia itself might have been disintegrated or hampered as a result of extended periods of invasion by male killing strains of *Wolbachia*. A protein of *Wolbachia*, Oscar’s, was manifested to engage with Masc through its Ankyrin repeats. The embryonic expression of Oscar’s has been revealed to hamper the masculinization trigger of Masc in two Lepidoptera insects: *Ostrinia furnacalis* and silkworm (*Bombyx mori*) resulting in the death of males [157]. Male death caused by *Wolbachia* is accomplished through a number of alterations besides other reproductive abnormalities occurring in Lepidoptera and Drosophila hosts involving the Oscar protein and discovery of candidate gene wmk (WO-moderated killing).

## 5. The Function of Scale Insect Companion Species in *Wolbachia* Host Fluctuations

The present study, using the superfamily Coccoidea as a model system, reveals the dynamics of *Wolbachia* in scale insects by exploring the complicated world of host switching. The super family Coccoidea (the scale insects) is distributed across the globe with >8200 species and 24 families [158,159]. Scale insects, just like numerous other Sternorrhyncha suborder members (including whiteflies, psyllids and aphids), feed exclusively on plants and are believed to be serious pests of agricultural crops [160]. They have been found in association with a large number of other arthropod species. Particularly, a number of these have been found in proximity with ants owing to trophallaxis (ants are attracted to the honeydew of scale insects) [161,162]. Although there are numerous similarities with other hemipterans, it has been reported that many of the species are anticipated to have small-to-moderate prevalence of *Wolbachia* [26] contrary to a u-shaped (majority of the species have very high or very low prevalence) predicted distribution reported for other groups [67]. The positive interaction between ants and their associated *Wolbachia* strains infecting scale insects also indicates a credible transfer route [26].

These preclusive data reveal a wider view of the dynamics of *Wolbachia* infection in scale insects and hence encourages us to explore strain diversity of *Wolbachia* and henceforth host-switching in scale insects. Another probable justification for having a weak impact of host geographical distance on *Wolbachia* sharing is host-switching through ecological vectors. These vectors may cause either temporary or permanent infection and transfer it to a far geographical place to recipient species from the donor species. There are many indirect or direct pathways that may act as the routes of transfer of *Wolbachia* from prey-predator [36,163] to host-parasitoid [164,165] and trophallaxis interactions [166]. Familiarity of physical direct interactions of scale insects and ants may furnish a mean for exchange of microbes as observed in other hemipteran groups [167,168]. Furthermore, it has been observed that credible means of horizontal transfer of Cardinium do exist between parasitoid and scale insects [169]. These ecological routes may have been used by *Wolbachia* for its spread within the communities of scale insects. The positive association of *Wolbachia* infection within members of scale insects and the associates exhibited that the ants might have a role in host-switching [170]. This innovative research provided the initial data about the strains of *Wolbachia* in scale insects and the noteworthy frequency of confected samples. It is interesting that *wSph 1* was recorded as the most frequently observed strain of *Wolbachia* in scale insects. In addition, it was observed that distance effect of host phylogeny is very important for promoting host switching in scale insects [171]. These findings help to unveil the complicated interactions observed among the scale insects.

## 6. The Use of *Wolbachia* in Leafhopper Vectors to Regulate Plant Hopper Populations

The plant hoppers are among the most threatening and destructive pest of rice. They suck the sap and oviposit the tissues of rice, and hence are a serious threat to rice production. Besides infestations of rice crop, these plant hoppers are also the vectors of various important plant viruses, including rice black-streaked dwarf virus, southern rice black-streaked dwarf virus rice stripe virus, grassy stunt virus, and rugged stunt virus [172]. The successful control of mosquitoes through *Wolbachia* prompts the application of a similar approach to manage plant hoppers. The sugarcane white leaf disease caused by a phytoplasma is also vectored by leafhopper *Yamatotettix flavovittatus* Matsumura (Hemiptera: Cicadellidae) [41]. It is believed to be among the most serious issues of sugarcane production resulting in significant yield losses in many sugarcane cultivating areas of Southeast Asia [42,43]. In natural populations of leafhoppers *Wolbachia* infection has been reported with high frequencies (>80%) [44]. Furthermore, the reproductive incompatibility along with the immaculate vertical transmission in leaf hoppers were caused after infection with *Wolbachia* [45]. The results stresses on the probability of future prospects of management of leafhoppers using *Wolbachia*. Research on the quantity of *Wolbachia* seeks to gain insight on the relationship of a symbiont with its hosts. To investigate the number of *Wolbachia*, the wsp gene was chosen to express the surface protein and measure *Wolachia* through qPCR. On the basis of the fact that a single copy of the wsp gene is present in each of the genomes, its absolute copy number was measured to predict the number of *Wolbachia*. The same method has been employed to determine the population of *Wolbachia* in many arthropod hosts [173,174,175]. *D. citri*, the psyllid, the population density of *Wolbachia* was raised during the succeeding nymphal instars [176]. The adults and nymphs of leafhopper *Wolbachia* concentrated and localized in bacteriomes [175] and hence, its multiplication was hampered by the available space in cells or the tissues being invaded. The latest research findings revealed that *Wolbachia* furnishes beneficent impacts to BPH (Brown Plant hopper). There is a high rate of egg production in BPH females infected with *Wolbachia* compared to uninfected females.

Nevertheless, the lifespan of BPH infected with *Wolbachia* is quite shorter than the uninfected BPHs, which might also explain the production of a very high number of eggs and lesser prevalence of the symbiont in the wild type BPH. Analogous to the BPH, *Wolbachia* also significantly enhanced the fecundity of SBPH (Small Brown Plant Hopper), which may also be affiliated with the increased number of Ovarioles containing mitotic germ cells and apoptotic nurse cells [177,178]. Additionally, *Wolbachia* impacts the expression of miRNA in SBPH to manipulate the gene expression associated with fecundity [179]. More genomic and experimental testimonies established that *Wolbachia* improves the fecundity of SBPH and BPH females through synthesis of key nutrients riboflavin and biotin [180]. This symbiont is an encouraging tool to devise efficient management practices against important agricultural pests, as is evident from the divergent impacts it has on arthropod vectors including leafhoppers (*Yamatotettix flavovittatus*) and plant hoppers of rice. The detrimental impacts range from ideal vertical transfer and incompatibility of the reproduction system to impacting fecundity via the synthesis of nutrients.

## 7. Future Prospective and Challenges

The control strategies of *Wolbachia*-based systems have been shown to be effective, and their application might lead to a drastically different and dynamic environment. Decoding the complex of changing aspects of *Wolbachia* in a variety of insect orders, such as dipterans, lepidopterans and hemipterans, is one major area of research. It examines control strategies, with a focus on the incompatible IIT and SIT. To overcome obstacles and efficiently manage *D. suzukii* populations, it is recommended that SIT be included in area-wide management and that the contingency plan of merging SIT/IIT be taken into consideration. A potential new approach that combines these two methods has been created and is presently being evaluated in open-field experiments against populations of *Aedes* mosquitoes.

This has indicated the necessity of funding efficient control measures. Biological control vector techniques are one of these acts, and *Wolbachia* has proven to have beneficial potential in the agriculture field. The efficacy of *Wolbachia*-based control relies on maternal transmission and induced cytoplasmic incompatibility, ensuring rapid spread through the pest population. Despite its success in mosquito control, progress has extended to protecting plants from rice pests. This synthesis explores *Wolbachia* phenotypic effects in mosquito control and its interactions with rice pest plant hopper, aiming to bridge findings from mosquito programs to potential applications in plant hopper control.

*Wolbachia* are ubiquitous endosymbiotic bacteria that are highly prevalent, especially in arthropods, because they can generate a variety of phenotypes in their hosts and move between host species with ease. Although the phenotypic impacts have been better understood, there is still much to learn about the processes underpinning *Wolbachia* host-shifting. A significant incidence of co-infected samples was shown by a recent study on the variety of *Wolbachia* strains in scale insects, with a particular emphasis on the common wSph1 strain. Produce that the host’s evolutionary distance was a significant factor affecting the host-shifting behaviour of scale insects, suggesting possible transmission pathways and possible host changes. This highlights the significance of ecological connectedness in the complex dynamics of *Wolbachia* host-shifting and lays the groundwork for further worldwide research investigating similar phenomena. The research’s suggested technique may be applied to larger datasets, providing information on the variables affecting *Wolbachia* host changes in various environments.

Maternally inherited endosymbionts, known as *Wolbachia,* are ubiquitous in the animal kingdom and exhibit amazing mutualistic and parasitic adaptations. A centennial investigation shows *Wolbachia* global abundance in nematodes and arthropods, establishing it as a key paradigm for studying symbiosis and fighting illnesses in humans and agriculture. The host range, phylogenetic diversity, genomics, and medical uses of *Wolbachia* are all included in this synthesis. The importance of the mobilome, in particular phage WO, for reproductive phenotypes is discussed. The system is a shining example of science education that is focused on discovery. Manuals on symbiosis may change as a result of recent developments in multiomics, gene functional assays, and human health applications. The molecular underpinnings of reproductive parasitism, genetic engineering for reductionist research, and the influence of *Wolbachia* on human illnesses and speciation are among the concerns of the future. The coming century promises revolutionary advancements, emphasizing the evolution of *Wolbachia* from a scientific curiosity to a model for microbial symbiosis and showing how fundamental science contributes to beneficial outcomes in education and translational research.

## 8. Conclusions

In conclusion, the influence of *Wolbachia* on evolution is highlighted by its ability to manipulate insect sex determination systems, as shown by the new protein Oscar. *Wolbachia* prevalence in Lepidoptera indicates predictability based on ecological parameters, with a latitudinal gradient and regional variance. The commercial fruit pest *Drosophila suzukii* may be managed with the use of the IIT and SIT. The ability of *Wolbachia* to suppress pests is dependent on both cytoplasmic incompatibility and maternal transmission. Using *Wolbachia*-based methods to protect plants from pests goes beyond only controlling mosquitoes. Molecular biology, ecology, and public policy must be combined by interdisciplinary teams in order to fully realise the potential of *Wolbachia*-based therapeutics.

## Figures and Tables

**Figure 1 biology-13-00151-f001:**
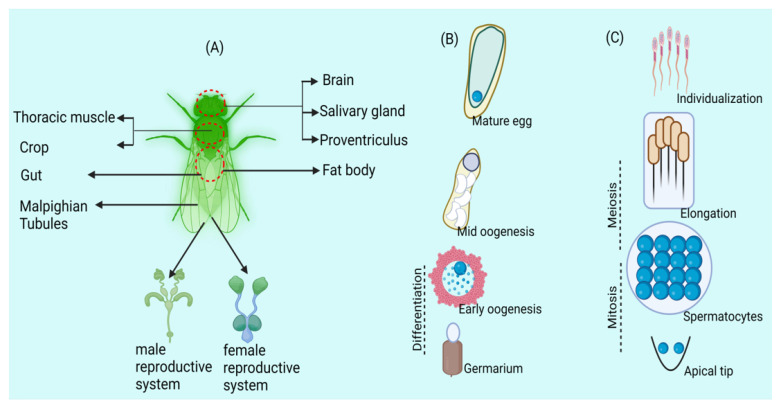
*Wolbachia* exhibits tissue tropism in arthropods, influencing germline stem cells. (**A**) The arthropod host Drosophila has *Wolbachia* labelled in its somatic and reproductive tissues. (**B**) Wolbachia in *Drosophila oogenesis*: Germline stem cells host infection, influencing egg chamber formation. Microtubules aid oocyte entry; posterior localization persists in mature eggs. (**C**) *Wolbachia* in *Drosophila spermatogenesis*: Germline stem cells carry infection, mitotic divisions yield *Wolbachia*-uneven spermatocytes, with individualization discarding *Wolbachia* into a waste bag.

**Figure 2 biology-13-00151-f002:**
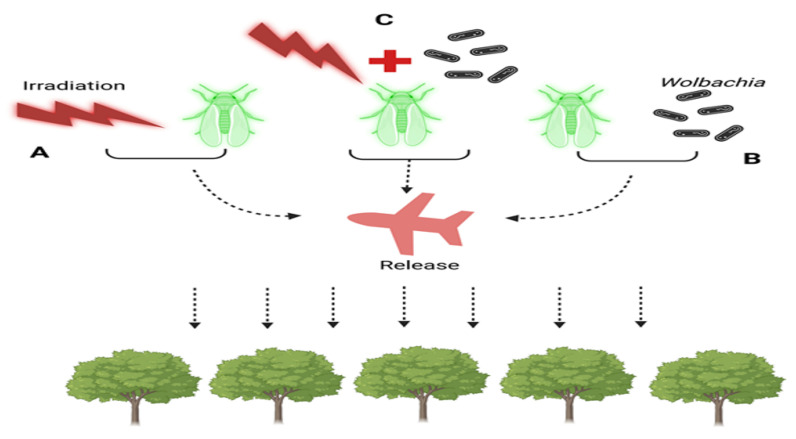
Strategies for Population Control in Sterile Insect Techniques (SIT) and Incompatible Insect Techniques (IIT): (**A**) Sterile Insect Technique (SIT): Males undergo sterilization through irradiation. (**B**) Incompatible Insect Technique (IIT): Males are sterilized through *Wolbachia* trans-infection. (**C**) Combination of SIT and IIT: male sterility is achieved through both irradiation and *Wolbachia* infection. In all three cases (**A**–**C**), sterilized males are released into the field to target and sterilize wild females of the population.

**Table 1 biology-13-00151-t001:** A thorough explanation of the host impacts and ways that *Wolbachia* spreads throughout several insect species in a range of settings and interactions.

Insect Name	Transmission Methods	Effect on Host	Citation
Rice Plant hoppers	Horizontal	The physiological functions of the host are impacted in observable ways by *Wolbachia*’s horizontal transmission, suggesting a complicated interaction between the bacterium and its host.	[73,74]
Wasps	Horizontal	Due to their dual function as possible immune modulators and vectors of horizontal transmission, parasitic mites present important questions in the context of predation.	[75,76]
Fruit Flies	Horizontal	Successful horizontal transmission and possible fitness impacts are revealed by hybridization with parasitic wasps, underscoring important dynamics in the complex host–parasite interaction.	[29,77]
Trypetids	Horizontal	The hybridization process with parasitoid wasps exhibits horizontal transmission and has the capacity to influence the reproductive characteristics of the host.	[78]
Psyllids	Horizontal	Feeding on common plants permits horizontal transmission and so offers the possibility of affecting the dynamics of host fitness.	[79]
Moths	Horizontal	Hybridization highlights the complex dynamics in the setting of interspecies interactions by revealing horizontal transmission and its possible implications on mating behaviour.	[24]
Ladybirds	Horizontal	Because predators make the spread of disease easier, possible consequences on host fitness in ecological dynamics should be taken into consideration.	[80]
Mites	Horizontal	Predation promotes horizontal transmission and may have an impact on the dynamics of mite populations in complex ecological interactions.	[81]
Various Diptera Insects	Vertical	Vertical transmission is demonstrated by transovarial transmission, which may have an impact on the kinetics of host reproduction.	[82]
Drosophila	Vertical	Actin-mediated transovarial transmission: *Wolbachia* affects host reproductive organs; comprehensive understanding of actin-mediated transmission mechanisms	[83,84]

**Table 2 biology-13-00151-t002:** An Extensive Review of *Wolbachia* Action Mechanisms and Their Effects on Host Physiology.

Mechanism of Action	Impact on Host Physiology	Citations
*Wolbachia* exhibits obligatory reproductive parasitism	*Wolbachia* has a complex effect on the physiology of its host species, depending on its population density, which is highly regulated by dietary variables and ambient temperatures.	[108,109]
Temperature-induced changes affect cytoplasmic incompatibility (CI) capacity	Modifies the ability for cytoplasmic incompatibility (CI), resulting in a complex reaction where some males retain incompatibility while females lose this reproductive characteristic.	[110]
reduced life expectancy, decreased viral replication, and changed vector competency	*Wolbachia* has a complex effect that includes preventing virus replication, shortening mosquito lifespans, and impairing mosquito vector competence.	[56,111]
Reduction in saliva production affecting feeding capacity	*Wolbachia* reduces salivary flow, which limits mosquitoes’ ability to eat.	[112,113]
Provocation of changes linked to lead resistance, iron homeostasis, RNA and DNA processing, and digestive enzymes	Iron homeostasis, RNA and DNA functions, digestive enzymes, and lead resistance are all altered by *Wolbachia*.	[114,115,116,117,118]
Hypotheses on action: Competition or Modulation of lipid production	Two possible ways that *Wolbachia* may use to regulate lipid synthesis are competition and modulation, respectively.	[119,120]
Regulation of cellular autophagy and pre-activation of immune system	*Wolbachia* stimulates genes linked to the Toll pathway, pre-activates the immune system, and modifies cellular autophagy.	[121]

## Data Availability

Data sharing is not applicable to this article.

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
