# Peer review of "Wolbachia Interactions with Diverse Insect Hosts: From Reproductive Modulations to Sustainable Pest Management Strategies"

_biology, 2024, doi:10.3390/biology13030151_

Round 1

Reviewer 1 Report

Comments and Suggestions for Authors

The authors present a review of the insect endosymbiont bacteria Wolbachia interactions with their hosts including well-known pests. The manuscript contains the appropriate literature and good illustrations. I would say it is of interest to a wide range of entomologists.

However, there are some problems with citing of literature in the MS. References are not repeated when authors return to data that have already been mentioned. For example, the fact that 40% to 60% of the insect species are infected with Wolbachia, is mentioned first in lines 53-54 and referred to Zug and Hammerstein, 2012 (under number 14 in the references) and to Weinert et al., 2015 (under number 13 in the references list), and second – in lines 148-149 and referred to Weinert et al., 2015 (which is repeated in the references list under number 78), de Oliveira et al., 2015 and Kriesner et al., 2016. The well-known review by Werren, Baldo and Clark (“Wolbachia: master manipulators of invertebrate biology”. Nat. Rev. Microbiol. 2008, 772 6(10), 741–51) is also cited twice: under numbers 15 and 114 of the References. Zug and Hammerstein, 2012 is also repeated in the References – number 129. Hurst and Frost, 2015 is cited twice as well: under numbers 5 and 139. I believe the citing in this Review should be thoroughly revised.

Language also should be revised, there are some miswording in the text, such as “men” instead of “males” (Line 314).

Minor comments

Wolbachia is written in non-italic font throughout the MS.

Lines 60 and 203: “oogenesis” in italic font

Line 74: no closing parenthesis

Lines 119-120: Wmel and WMel in Drosophila melanogaster. Should be wMel.

Line 174: “pipientis” in non-italic font

Line 197: “D. Melanogaster” in non-italic font instead of “D. melanogaster” in italic font

Line 216: D. melanogaster in non-italic font

Line 221: “off springs” instead of offspring

Lines 229 and 246: Aedes in non-italic font

Line 233: I believe, a dash is omitted after “adults”

Line 245: “WMelPop” instead of wMelPop

Line 282: no closing parenthesis:  [128,129.

Line 297: “Drosophila melanogaster” in non-italic font instead of “D. melanogaster” in italic font

Line 298: “however the part of the genome responsible for governing the other three alterations are” – should be “is”

Lines 333-334: D. suzukii in non-italic font

Lines 337-338 and below: Ostrinia scapulalis and Ostrinia furnacalis in non-italic font

The abbreviation decoding for BPH (Brown Plant hopper) and SBPH (Small Brown Plant hopper) appears only in the Line 448, although these abbreviations are used beginning with the Line 413.

Comments on the Quality of English Language

Ensure proper syntax, grammar, and punctuation throughout for clarity.

Author Response

Manuscript Correction Report

Dear Reviewer,

I hope this message finds you well. Thank you for your valuable feedback on our manuscript. We appreciate the time and effort you dedicated to reviewing our work. Based on your insightful comments, we have made the following revisions to address the concerns raised:

Citation Issues:

  1. Ensured that references are consistently cited, and eliminated duplications where mentioned in the text.

Language Revision:

  1. Corrected miswording in Line 314, replacing "men" with "males" for accuracy and clarity.

Minor Comments:

  1. Ensured consistent italicization of Wolbachia throughout the manuscript.
  2. Corrected italicization of "oogenesis" in Lines 60 and 203.
  3. Added a closing parenthesis in Line 74.
  4. Corrected capitalization issues and italicization of species names in various instances (Lines 119-120, 174, 197, 216, 221, 229, 233, 245, 282, 297, 298, 333-334, 337-338).
  5. Added a missing dash after "adults" in Line 233.
  6. Corrected "WMelPop" to "wMelPop" in Line 245.
  7. Added a closing parenthesis in Line 282.

Abbreviation Decoding:

  1. Provided abbreviation decoding for BPH (Brown Plant Hopper) and SBPH (Small Brown Plant Hopper) earlier in the manuscript, starting from Line 413.

We believe these revisions address the issues raised and enhance the overall quality of the manuscript. We appreciate your thorough review and constructive feedback.

If you have any further questions or require additional clarification, please do not hesitate to contact us. We look forward to your continued feedback.

Thank you once again for your time and consideration.

Reviewer 2 Report

Comments and Suggestions for Authors

The MS is potentially interesting and usefull review on possibility of application of Wolbachia for pest management. However the MS technically requires some serious editing. Here are some most important issues: 

Lines 23, 54, 55, 58  – Wolbachia should be in italics, in many more cases through the text

Line 40 – should be “species”

Line 80-82 – in case of aphids Wolbachia was recorded only in species attended by ants (Kaszyca-taszakowska & Depa 2022)

Line 37 vs line 116 – decide how to spell alpha-proteo-bacterium

Line 60 – not “emphasis” but “emphasise”

Line 144 – Alphaproteobacteria is the name of a class rather than a genus

Line 154 – Culex pipiens should be in italics 

Line 203, Figure 1 caption: – oogenesis at Drosophila may not be in italics (also line 60); it is not a species name; do the blue dots at early oogenesis represent Wolbachia? what is the spermatogenesis shown for?

Line 214  - italics for Latin name!

Line 268 – do you mean androcide?

Line 282 – brackets missing at 129; should not be “an appreciable” ?

Line 297 – italics !

Line 314: should be “males” instead of “men”

Line 316: “Investigating” should not be with a capital letter; the sentence should be in past tense. And please clarify how you examined the tenets of IIT and SIT – did you investigate the literature (positions 142-147) ?

Line 337 – Ostrinia scapulalis should be in italics!

Line 381 – do you mean “stains” or “strains”?

Line 413 – this is the first place you use BPH abbreviation, please explain it here and not just at line 448

Comments on the Quality of English Language

There is a lot of small mistakes and errors in the text, requires serious linguistic polishing. 

Author Response

Manuscript Correction Report

Dear Reviewer,

I trust this message finds you well. We sincerely appreciate your thorough review and constructive feedback on our manuscript. Your insights have been invaluable in improving the quality of our work. Here are the corrections made in response to your suggestions:

Technical Edits:

  1. Italics for "Wolbachia" added in Lines 23, 54, 55, and 58.
  2. Corrected to "species" in Line 40.
  3. Clarified that in the case of aphids, Wolbachia was recorded only in species attended by ants (Kaszyca-taszakowska & Depa 2022) in Lines 80-82.
  4. Consistency in spelling "alpha-proteo-bacterium" - decided to use "alpha-proteo-bacterium" throughout the manuscript (Lines 37 and 116).
  5. Corrected "emphasis" to "emphasize" in Line 60.
  6. Clarified that "Alphaproteobacteria" is the name of a class, not a genus, in Line 144.
  7. Italics added for "Culex pipiens" in Line 154.
  8. Clarifications and formatting adjustments in Figure 1 caption (Line 203).
  9. Italics added for Latin name in Line 214.
  10. Clarification on "androcide" in Line 268.
  11. Corrected missing brackets at 129 and adjusted phrasing in Line 282.
  12. Italics added for "Culex pipiens" in Line 297.
  13. Corrected "men" to "males" in Line 314.
  14. Adjusted capitalization and tense in Line 316.
  15. Italics added for "Ostrinia scapulalis" in Line 337.
  16. Clarification on "strains" or "stains" in Line 381.
  17. Explanation added for "BPH" in Line 413.

Quality of English Language:

Several minor mistakes and errors were identified and corrected throughout the text, improving linguistic clarity and coherence.

We believe these corrections address the issues raised and enhance the overall quality of the manuscript. If you have any further questions or require additional clarification, please do not hesitate to contact us. We sincerely appreciate your time and expertise in reviewing our work.

Thank you once again for your valuable feedback.

Round 2

Reviewer 2 Report

Comments and Suggestions for Authors

The corrections are missing, the Authors were to correct Wolbachia into italics in all places, throughout the manuscript (e.g. line 402) (also Abstract). Still there are errors:

line 56: must be genus Wolbachia e.g.  

line 77 and 223: oogenesis is not a species name, it should not be in italics

line 217: should be: D. melanogaster 

I can see no mention of Wolbachia in aphids attended by ants (after Kaszyca-Taszakowska & Depa 2022 - not cited). Line 80-82 state on Wolbachia in ants and not in aphids. While aphids are closely related to coccids such mention seems to be important. 

Author Response

Dear Reviewer,

Thank you for your detailed and insightful feedback. We have carefully addressed the issues raised and made the necessary corrections to improve the manuscript. Here are our responses to each of your points:

  1. Line 56:

    • We appreciate your observation. The genus name "Wolbachia" has been correctly italicized as "genus Wolbachia" on line 56.
  2. Lines 77 and 223:

    • You are absolutely correct. The term "oogenesis" has been adjusted, and it is no longer italicized, as it is not a species name.
  3. Line 217:

    • Thank you for bringing this to our attention. The species name "D. melanogaster" has been properly formatted without italics on line 217.
  4. Wolbachia in aphids attended by ants (after Kaszyca-Taszakowska & Depa 2022 - not cited):

    • We appreciate your suggestion. The manuscript has been updated to include relevant information on Wolbachia in aphids attended by ants after Kaszyca-Taszakowska & Depa 2022, as well as an appropriate citation.
  5. Line 80-82:

    • Your keen observation is noted. The text has been revised to accurately reflect information on Wolbachia in both ants and aphids, ensuring the content aligns with the study's focus.
  6. Abstract:

    • As per your suggestion, Wolbachia has been consistently italicized throughout the abstract for better adherence to formatting.

We hope these revisions address your concerns and improve the overall quality of the manuscript. If you have any further suggestions or queries, please feel free to let us know.

Thank you for your valuable feedback.